# Using Foundation Models to Promote Digitization and Reproducibility in Scientific Experimentation

**Amol Thakkar**
IBM Research Europe – Zurich
Switzerland
tha@zurich.ibm.com

**Andrea Giovannini**
IBM Research Europe – Zurich
Switzerland
agv@zurich.ibm.com

**Antonio Foncubierta**
IBM Research Europe – Zurich
Switzerland
fra@zurich.ibm.com

**Carlo Baldassari**
IBM Research Europe – Zurich
Switzerland
carlo.baldassari@zurich.ibm.com

**Dimitrios Christofidellis**
IBM Research Europe – Zurich
Switzerland
dic@zurich.ibm.com

**Federico Zipoli**
IBM Research Europe – Zurich
Switzerland
fzi@zurich.ibm.com

**Gianmarco Gabrieli**
IBM Research Europe – Zurich
Switzerland
gga@zurich.ibm.com

**Jannis Born**
IBM Research Europe – Zurich
Switzerland
jab@zurich.ibm.com

**Mara Graziani**
IBM Research Europe – Zurich
Switzerland
mgr@zurich.ibm.com

**Marvin Alberts**
IBM Research Europe – Zurich
Switzerland
marvin.alberts@zurich.ibm.com

**Matteo Manica**
IBM Research Europe – Zurich
Switzerland
tte@zurich.ibm.com

**Michael Stiefel**
IBM Research Europe – Zurich
Switzerland
michael.stiefel@zurich.ibm.com

**Oliver Schilter**
IBM Research Europe – Zurich
Switzerland
oli@zurich.ibm.com

**Patrick W. Ruch**[*]
IBM Research Europe – Zurich
Switzerland
ruc@zurich.ibm.com

**Teodoro Laino**
IBM Research Europe – Zurich
Switzerland
teo@zurich.ibm.com

[*]Corresponding author.

NeurIPS 2023 AI for Science Workshop.

## Abstract

Accelerating scientific discovery through AI relies on the availability of high-quality data from scientific experimentation. Yet, scientific experimentation suffers from poor reproducibility and data capture challenges, mostly stemming from the difficulty in transcribing all details of an experiment and the different ways in which individuals document their lab work. With the emergence of foundation models capable of processing multiple data modalities including vision and language, there is a unique opportunity to redefine data and metadata capture and the corresponding scientific documentation process. In this contribution, we discuss the challenges associated with lab digitization today and how multi-modal learning with transformer-based architectures can contribute to a new research infrastructure for scientific discovery in order to fully describe experimental methods and outcomes while facilitating data sharing and collaboration. We present a case study on a hybrid digital infrastructure and transformer-based vision-language models to transcribe high-dimensional raw data streams from non-invasive recording devices that represent the interaction of researchers with lab environments during scientific experimentation. The infrastructure is demonstrated in test cases related to semiconductor research and wet chemistry, where we show how vision-language foundation models fine-tuned on a limited set of experiments can be used to generate reports that exhibit high similarity with the recorded procedures. Our findings illustrate the feasibility of using foundation models to automate data capture and digitize all aspects of scientific experimentation, and suggest that the challenge of scarce training data for specific laboratory procedures can be alleviated by leveraging self-supervised pretraining on more abundant data from other domains.

## 1  Introduction

AI has been used extensively to support scientific discovery [1], such as in molecular discovery [2], to accelerate and enhance simulations [3], support lab automation and robotics [4], or to detect patterns in experimental data [5]. However, today there still exists a substantial gap between the generation of experimental data and the use of that data to train and validate AI models. First, data is typically not created in a form that is readily usable for training machine learning models. A significant amount of time has to be spent on data cleaning and preparation, in some cases estimated to be over 60% of the work by data scientists [6, 7]. Second, experimental data has the fundamental challenge that it tends to be derived from poorly reproducible experimentation. It has been estimated that up to 70% of experimentation is not reproducible because of flawed or missing data and metadata, while experimental replicates confirm the original findings in only one-third to one-half of cases [8, 9]. In a survey by *Nature* polling over 1500 researchers, more than half of the respondents stated that they have failed to reproduce even their own experiments [10]. We postulate that the lack of high-quality data, needed to unlock the full potential of AI for science, arises from legacy practices leading to cumbersome documentation and inadequate tooling during scientific experimentation. With foundation models emerging in recent years that can process observed inputs on par or exceeding human ability, we draw attention to the intriguing and unprecedented opportunities that those models offers for digitising operations in scientific discovery, while pointing out related challenges and pitfalls.

The emergence of vision transformers has brought the advantages of self-supervised pretraining on large-scale unlabelled datasets from the language to the image domain [11] and multi-modal learning [12, 13]. Girdhar *et al.* introduced the Action Transformer model to recognize actions performed by a person based on the context of the scene and person-specific self-attention [14]. For transcription of surgical procedures, Kiyasseh *et al.* reported a machine learning system leveraging a vision transformer and contrastive learning to decode sequences comprising three surgical subphases and ten different discrete surgical gestures from videos recorded during robotic surgical procedures [15]. The work demonstrated the capability to identify the surgical subphase, the type of gesture, and an assessment of the skill level (low or high) when analyzing videos across different surgeons and hospitals. The results of the system may be used to document procedures, provide postoperative feedback to surgeons, and optimize for desirable patient outcomes. Further, Ferreira *et al.* described transformer-based video segmentation for fine-grained classification of 19 exercise phases and count-

ing of repetitions to quantify and provide feedback on personalized workout routines [16]. Generally, transformer-based architectures for featurization and sequence prediction exhibit advantages in terms of accuracy and runtime over state-of-the-art human action recognition methods from videos [17]. To benefit from the rich representation learning of transformers and increasing performance with scale, pretraining on large quantities of data and subsequent fine-tuning on domain-specific actions is highly desirable [18, 19]. However, for specialized human tasks, such as those involved in scientific experimentation, training data is notably scarce and techniques such as transfer learning and domain adaptation are promising approaches worth exploring.

After summarizing the current state of lab digitization (Section 2), this work describes two novel contributions toward the development of AI-generated transcriptions for scientific experimentation: a hybrid architecture for multi-modal data capture from both scientific instruments and users (Section 3), and the first vision-language foundation model to demonstrate the automatic documentation of laboratory procedures (Section 4). We tackle the data scarcity problem related to videos of laboratory procedures by leveraging unsupervised learning on abundant video data from other domains for pretraining, and demonstrate the applicability of our approach in two fields of scientific experimentation: semiconductors and wet chemistry. The current feasibility and future potential of using foundation models to transcribe complex laboratory procedures is discussed (Section 5) and we conclude our contribution with closing remarks on recommendations and opportunities in this nascent field (Section 6).

## 2 Digitizing the lab: Opportunities and challenges

### 2.1 Data capture

The last three decades have seen the transition from paper based documentation to electronic lab notebooks (ELNs) and laboratory information management systems (LIMS) [20, 21, 22, 23]. However, despite their long history, their adoption in research laboratories is lagging [20, 24, 25]. A survey conducted by Kanza *et al.* [21] describes a fragmented market comprising over 70 active ELN products. Among the various factors cited, major concerns lay in the disconnect between the experimental record, the data, and the software/devices used to generate them [22]. This partly arises from the poor automation and integration between the devices that generate data and the means of documentation. For instance, while ELNs support upload of raw data, the onus is on the human operator to determine which data was generated from a given experiment, obtain the data from the instrument, and upload it to the corresponding entry in the ELN.

### 2.2 Data sharing

Furthermore, for data that is collected and documented, concerns have been raised around data sharing and reuse [26, 23]. Wilkinson *et al.* have outlined a set of guidelines describing the need to improve data infrastructure supporting reuse, known as the FAIR Data Principles [26]. They highlight the goal is not data management in itself, but rather as a key driver to support knowledge discovery and innovation. The principles have now been extended to research software [27], and have started to see adoption across the community, from funding agencies, to academia, and industry [28]. However, there remains much work to be done to develop the infrastructure necessary to support their successful implementation. As such there are plentiful opportunities to develop the infrastructure to support data integration and capture, as well as new methods for the seamless documentation of experimentation.

### 2.3 Laboratory virtualisation and user interaction

Beyond data capture, another opportunity that exists in the digitisation of laboratory environments is their virtualisation. Virtual laboratories encompass concepts such as the digital twin and can be used to support education and manufacturing [29, 30, 31]. Advantages of virtual labs include a more accessible and immersive experience for learning and communication, while the introduction of augmented reality (AR) may facilitate the guidance of manual labor tasks in particular for complex processes. While AR has been applied commercially in warehousing and logistics [31], applications in scientific laboratories have so far not yet gained traction. One benefit of AR in labs that has received attention is the aspect of hands-free working. Here, voice-activated assistants have also been proposed in commercial systems as a means to simplify note-taking on-the-fly [32, 33]. As with the

aforementioned technologies for data capture, the main challenges to consider are the effort needed to obtain interoperability between such solutions and laboratory infrastructure, as well as the ability to make data readily available for analysis.

# 3 Hybrid and multi-cloud infrastructure for AI-enabled labs

The analysis of opportunities and challenges from previous efforts described in Section 2 allows us to define five pillars of requirements to address in support of an end-to-end AI-enabled pipeline for scientific experimentation:

1. Seamlessly digitize user actions.

2. Minimize local computations while fulfilling data governance constraints.

3. Incorporate foundation models.

4. Integrate data from devices with user actions.

5. Intuitively represent experiments in real time.

Based on these five pillars, expanded below, we designed our infrastructure, which is shown schematically in Figure 1A. The figure highlights edge computing for preparing multiple streams of data, such as video, and serving multiple models that can be hosted in different cloud environments, addressing data and model governance by using a hybrid multi-cloud approach.

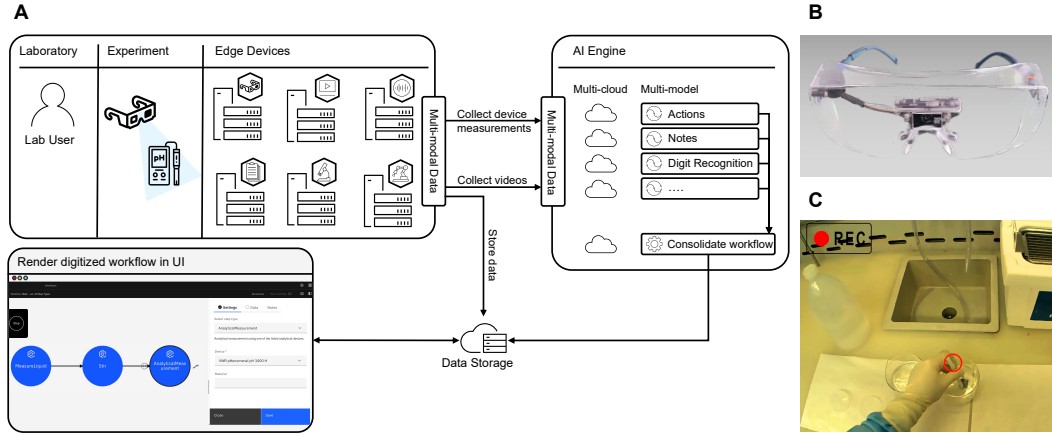

Figure 1: **An overview of the multi-modal, multi-model, multi-cloud infrastructure for digitising operational workflows.** (**A**) Multiple data streams are processed locally, then sent to the AI Engine. The AI Engine is composed of multiple models that are hosted in different cloud environments. The output of the multiple models is consolidated into a workflow by a last foundation model and stored. Finally the workflow is refreshed in the UI in realtime to show the last update in the workflow. (**B**) Lab safety glasses retrofitted with video capture devices for POV recording and gaze tracking. (**C**) Sample frame from a point-of-view recording of a stirring procedure during chip rinsing, including the instantaneous location of the researcher's gaze (red circle).

## 3.1 Seamlessly digitize user actions

To capture data seamlessly we integrated an off-the-shelf point-of-view (POV) video recording and eye tracking system into standard laboratory safety eyewear (Figure 1B). The eye tracking system streams the video recorded from the front camera together with the synchronized gaze to a local software component, which segments the streams into clips. Gaze tracking provides additional information about the user intentions in a non-intrusive manner. This additional signal is leveraged by the rest of the system.

### 3.2 Minimize local computations while fulfilling data governance constraints

To keep the local resources to a minimum, the majority of the computations are done in different cloud environments. The flexibility to choose cloud provider, data, and model locality, enables the enforcement of governance policies set out by the laboratory. In addition, the integration of edge computing devices across the network enables processing to be conducted at source minimising the risk of data leakage [34]. The local computations are limited to segmenting the video and gaze streams into clips, which are then uploaded to a cloud object storage followed by publishing an event per clip to the infrastructure broker.

### 3.3 Incorporate foundation models

The infrastructure supports both CPU and GPU clusters, the latter being used to serve foundational models. While the majority of the infrastructure is deployed on CPU only clusters, GPUs are triggered by the messages arriving to the infrastructure broker that relate to the use of foundation models. Use cases include inferring the description of the action performed from a video clip, estimating volume measured in a scaled cylinder from a video clip, recognizing digits in the videos, or improving the quality of voice notes. To support data and model governance policies, as well as achieve maximal flexibility the infrastructure follows a hybrid multi-cloud approach, thus can be configured on different cloud environments, ranging from on-premise to public clouds.

### 3.4 Integrate data from devices with user actions

The infrastructure is not limited to the collection of video and gaze streaming. Laboratory devices can also stream data to the infrastructure. Analogous to video and gaze, the device stream undergoes minimal local processing and the segmented data is sent to a cloud object storage or directly via the message payload to the infrastructure broker. The link between user actions and device output happens by triangulating the time of capture, the detected type of the action, and the device that is publishing data. The established link between device data and actions provides the required experimental context to trace back the conditions and the purpose of each experiment.

### 3.5 Intuitively represent experiments in real time

Beyond a backend able to capture and digitize the experiments in real time, a fundamental requirement is to present the result of the digitization in an intuitive manner. We built a user interface that refreshes in real time and shows the actions as nodes of a directed acyclic graph (DAG) (Figure 1A). By clicking on each node a widget opens on the side and shows configuration options and data, which are specific for each type of action. In addition to showing the captured actions, we also allow the user to modify, delete, create and connect new nodes to the experiment, allowing for a hybrid mode, where part of the experiment is captured automatically, and part is corrected or added by the user.

## 4 Foundation models and multi-modal learning for data capture in scientific experimentation

### 4.1 Multi-modal foundation models

Since the inception of the Transformer block [35], we have witnessed a revolution that has impacted language modeling and beyond. The recent advances in Large Language Models (LLMs) training [36, 37, 38, 39] and the recent wave of foundation models for language being released has paved the way for their pervasive use in various scientific disciplines [40, 41, 42, 43, 44, 45, 46]. In parallel, the same paradigms diffused in the computer vision space with a series of seminal works that proposed effective strategies to build foundation models for images and videos [47, 48, 49] . More recently, a natural evolution has led the community to build holistic models able to combine multiple modalities to exploit the potential of foundation models that span the full perception horizon [13, 50, 51, 12, 52]. Multi-modal models enable embedding an intelligence layer in systems spanning sensory perception to accomplish arbitrary tasks in challenging conditions, such as tracking actions in complex environments like research laboratories.

## 4.2 A vision-language model for laboratory procedure digitization

To showcase the impact of multi-modal foundation models declined in the case of laboratory procedure digitization we devised an encoder-decoder vision language model (cf. Figure 2) that relies on a pretrained encoder for vision – a VideoMAE [49] backbone trained with tube masking on egocentric videos from the Ego4D dataset [53] – that is coupled with a pretrained GPT2 model [40] as a decoder. The full model is fine-tuned on videos collected using the prototype goggles introduced in Section 3 considering two laboratory setups: development of semiconductors and standard wet chemical procedures. In both applications, we fine-tune both the encoder and decoder components using the video stream as input and generating a caption describing the action performed by the user where the ground-truth is defined by crowd-sourced annotations. The captions are then processed and converted into a workflow including the predicted steps represented as a DAG (cf. Figure 2).

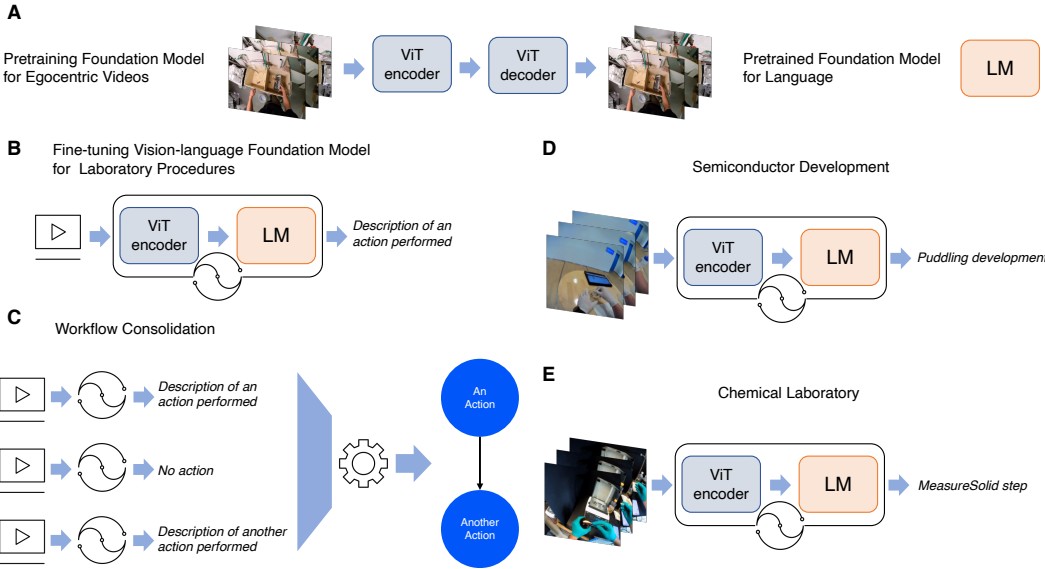

Figure 2: **Multi-modal foundation model for laboratory procedure digitization.** (**A**) We consider an encoder-decoder foundation model for egocentric videos we train on Ego4D [53] and a pretrained decoder-only language model from the GPT-family [40]. (**B**) Combining the vision encoder and the language decoder we are able to fine-tune both components for generating descriptions of the videos collected in a laboratory. (**C**) Model predictions are processed to discard non relevant actions and generate a DAG representing the executed procedure. (**D**) Collecting videos covering rinsing and three semiconductor development patterns: puddle, figure right and circular; we fine-tune the model for semiconductor development. (**E**) Collecting videos covering a simple recipe, such as extraction caffeine, we fine-tune the model for chemical laboratory procedure tracking.

**Dataset for semiconductor development.**    For the semiconductor use-case, 10 operators recorded 139 workflows for a total of 5799 videos (4 s clips at 4 Hz). In the recorded videos, there are 5 steps of interest: chip rinsing, figure eight development, circular development, puddle development as well as no action (more details in Appendix A).

**Dataset for chemical laboratory.**    For the chemical laboratory use-case, 51 operators in 2 different environments recorded 168 workflows for a total of 6877 videos (4 s clips at 4 Hz). In the recorded videos, 8 steps of interest have been considered: measure solid, analytical measurement, measure liquid, phase separation, stir, add, collect layer and no action (more details in Appendix A).

Table 1 reports the performance of the fine-tuned models in the two considered settings. The performance of the model is calculated comparing the workflow generated after consolidating the model predictions and the ground-truth annotations defined via crowd-sourcing. To measure the performance, we define a similarity measure between workflows computed using the normalized Levenshtein distance [54], where we identify each workflow with a sequence of characters mapped to

the steps considered and we measure the similarity between the sequences derived from the prediction and the ground-truth. In both cases, despite the limited number of experiments considered for model training (110 for the semiconductor case and 168 for the wet chemistry case), the model is able to reconstruct the workflow from the video recordings of the procedures: 0.861 average similarity in the semiconductor setting and 0.665 average similarity in the chemical procedures. The higher similarity score in the semiconductor case is explained by the lower complexity, directly linked to the reduced number of considered steps.

| Laboratory setup | Number of training experiments | Number of steps | Similarity score |
|---|---|---|---|
| Semiconductor | 110 | 5 | 0.861 |
| Chemistry | 168 | 8 | 0.665 |

Table 1: **Workflow reconstruction for the two considered laboratory setups: semiconductor and chemistry.** Besides reporting the number of training experiments and steps considered, we include the similarity score between generated and ground-truth workflows (considering a flattened text representation of the DAG) computed using the normalized Levenshtein distance [54].

Figure 3A depicts an exemplary workflow prediction exhibiting a high similarity score (0.889). In the chemical laboratory experiments (cf. Figure 3B), actions related to measurements and analytical instruments are tracked more accurately, while the model tends to confuse predictions related to operations involving the usage of similar glassware (e.g., *CollectLayer* and *PhaseSeparation*). In the case of semiconductor development (cf. Figure 3C), confusion tends to arise between actions that exhibit similar patterns at the frame rate considered (e.g., *Figure Eight development* and *Circular development*).

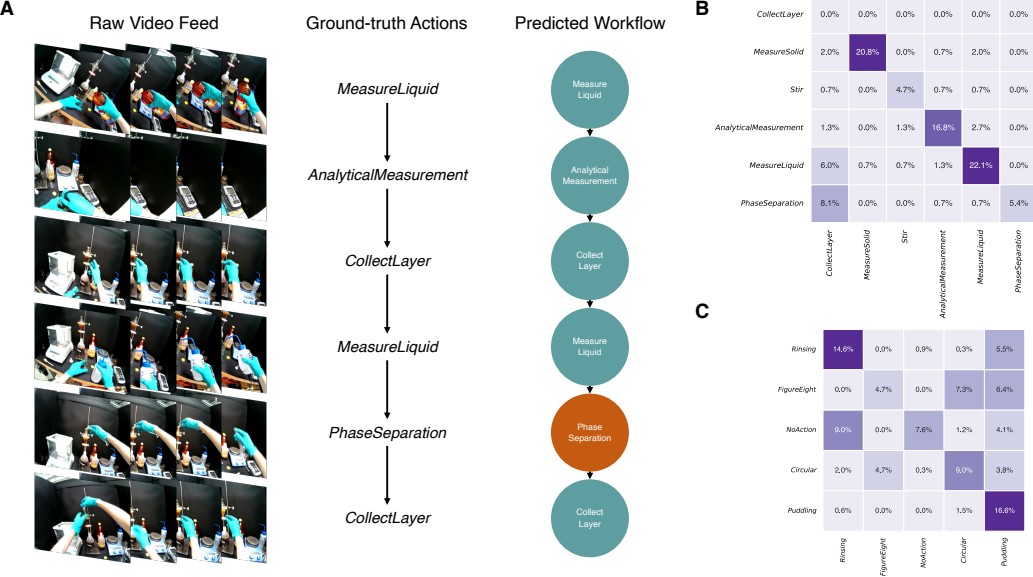

Figure 3: **Analysis of the workflow prediction pipeline.** (**A**) From left to right: raw videos recorded and streamed via the camera-equipped safety goggles, ground-truth steps associated to the executed procedure, predicted workflow using the vision-language fine-tuned in the specific lab setting (missing prediction highlighted in dark orange). (**B**) Confusion matrix for the chemical laboratory experiments excluding the *NoAction* step for clarity and the *Add* step as the latter is never predicted by the model (the complete matrix is reported in Appendix Figure 4). (**C**) Confusion matrix for semiconductor development experiments. Both confusion matrices are normalized over the number of test set videos related to the steps considered.

# 5 Discussion

The desire to facilitate and automate data and metadata capture during scientific experimentation is more prominent than ever. Besides improving reproducibility, collecting and sharing fine-grained experimental data related to procedures and outcomes is key to accelerate discoveries in the scientific community. With the emergence of large multi-modal models, the technology to directly digitize the parameters of manual experimentation is becoming accessible to a broad scientific user base. In combination with a layered computing infrastructure spanning from connected devices to edge computing to multi-cloud environments, we have shown how data generated by researchers through wearable devices and scientific tools can be readily captured and transcribed by vision-language models fine-tuned to describe experimental workflows.

Following first-pass workflow generation without any note-taking overhead to the experimentalists, the average similarity scores of 0.861 for semiconductor workflows and 0.665 for wet chemistry workflows indicate that useful protocols can be drafted on the basis of moderate domain-specific training efforts involving <200 training experiments for each workflow category (semiconductor and wet chemistry, respectively). Amending and augmenting the protocol at this stage is arguably a far lesser burden to researchers compared to deliberate documentation during experimentation. In particular, the association of video snippets with specific steps in the workflow allows any version of a particular action to be retrieved and reviewed at any point in time. Expanding the vocabulary and training scope of multi-modal foundation models for laboratory digitization promises to provide successively higher fidelity in transcribing even complex experimental procedures across various domains such as chemistry, biology, and physics.

Practical considerations related to the implementation of the proposed system should take into account the computational cost, storage requirements and sustainability through time. Strategies to consider in order to manage such challenges include model compression techniques such as distillation, the selection of appropriate pre-processing to reduce the data volume to be stored including gaze-based prompting, and storing appropriate multi-modal embeddings instead of raw data.

# 6 Concluding remarks

Multi-modal foundation models offer a new route to promote digitization of scientific experimentation. The fusion of laboratory data sources to provide a complete description of an experiment requires special attention on data interoperability, integration of heterogeneous devices, and a diverse population of researchers with individual preferences and habits. With the approach outlined in the present contribution, we have taken a first step in demonstrating the feasibility of fully automatic transcription of laboratory actions leveraging foundation models. Now, the full potential of the technology to become a valuable aid for scientific discovery is ready to be explored.

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

# A   Dataset details

| Laboratory setup | Training set | Validation set | Testing set |
|---|---|---|---|
| Semiconductor | 110 | 13 | 16 |
| Chemistry | 168 | 7 | 18 |

Table 2: Recorded workflows splitting for the two use-cases.

| Laboratory setup | Training set | Validation set | Testing set |
|---|---|---|---|
| Semiconductor | 5799 | 336 | 343 |
| Chemistry | 5911 | 301 | 665 |

Table 3: Recorded videos splitting for the two use-cases.

# B   Result details

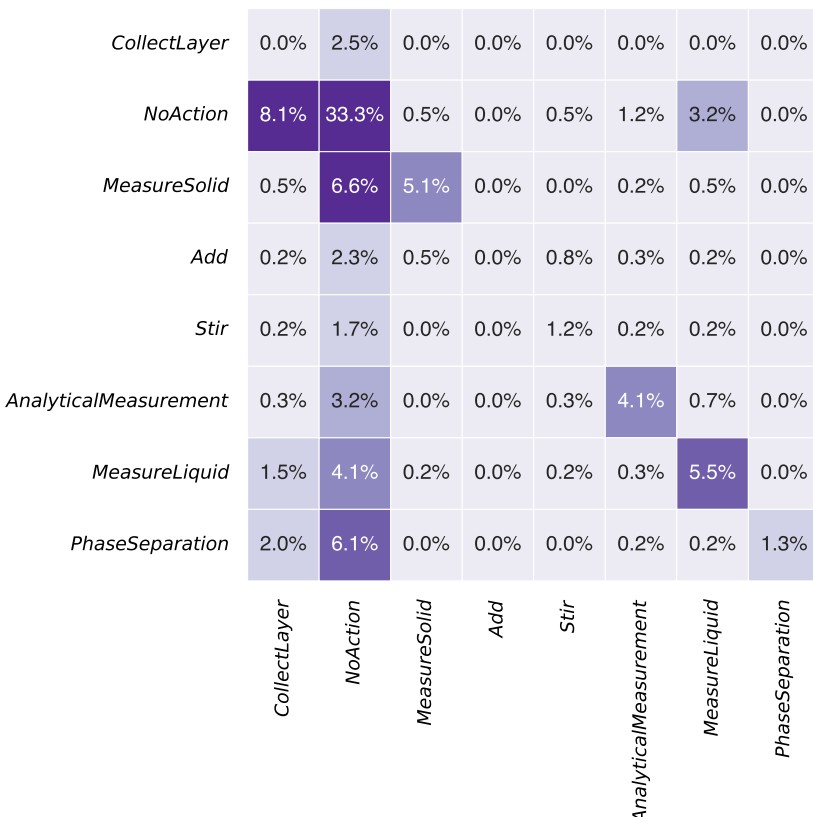

Figure 4: **Confusion matrix considering all steps for the chemical laboratory.** It is evident that the *NoAction* step covers the majority of the videos in the test set. Interestingly, *CollectLayer* steps are mostly predicted in relation to *NoAction* steps, most likely because of the presence of glassware and tools for layer collection at the center of the scene. The model is not predicting any *Add* steps as their duration is limited and they are not well represented in the recorded videos.

