# OpenReview forum: "Using Foundation Models to Promote Digitization and Reproducibility in Scientific Experimentation"
_NeurIPS.cc/2023/Workshop/AI4Science — NeurIPS2023-AI4Science Poster_

### Official Review · Reviewer_p8hg · 2023-10-15
**title_placeholder**

**Rating:** 6
**Confidence:** 3

**Review:**

Summary

This paper uses video captioning network to transcribe raw videos from recording devices into user actions conduted during scientific experiments. The network is fintuned using a small set of (vidoe, caption) pairs under two different laboratory settings, e.g., semiconductors and wet chemistry. Once finetuned, the network is capable of generating workflow that is a sequence of user actions with reasonable accuracy.


Strength

- The entire process, from video data capture to processing and automatic documentation, serves as a simple but complete demonstration of the potential usefulness of multi-modal models in automating experiment documentation.
- The research direction is promising, i.e., in employing versatile large multi-modal models for increasingly complex experimental procedures.
- Acquisition of data in realistic experimental setups is invaluable. It would greatly benefit the community if the authors could share the data acquired.
- Writing is overall clear and easy to understand.


Weakness

- The paper should refrain from claiming that the work is based significantly on foundation models. The authors simply fine-tunes a pre-trained network for video captioning. Their main contribution lies in making the entire data processing pipeline running that does not rely on specific properties of the foundational model.
- The proposed method does not scale well with the complexity of the range of possible user actions or experimental protocols because it relies on ground-truth user action annotations.

---

### Official Review · Reviewer_zknh · 2023-10-22
**Automation in chemistry laboratories.**

**Rating:** 7
**Confidence:** 1

**Review:**

The article presents a novel framework aimed at automating data capture during experiments in chemistry and material science labs. As a non-expert in the domain, I found the content intriguing, particularly admiring the ambitious vision that encompasses intricate vision tasks, language tasks, and a multi-modal transformer model.

The authors outline five essential components required for an automated laboratory and elaborate on how their framework addresses these "pillars". This work can be viewed as a significant stride towards achieving complete automation in chemistry labs.

I'd like to offer a few comments:

1) I recommend incorporating a concise introductory section that elucidates the primary motivation and the specific results achieved. This would greatly benefit readers from different domains (like myself), offering a clearer understanding right from the onset. I personally struggled a bit to extract this essence from the manuscript during the first time reading.

2) The paper doesn't explicitly convey the computational demands associated with developing this tool. Specifically, how extensive is the training process for the multi-modal transformer?

3) The universality of the framework remains a bit ambiguous. Can it be effectively deployed in labs outside of the authors' immediate environment, even if they're within the same domain? Furthermore, is there potential for its application in other domains, such as automated labs in biology or physics?

---

### Meta-Review · Area_Chair_Qwov · 2023-10-27

**Recommendation:** Accept (Poster)
**Confidence:** 5

**Metareview:**

**Overview:**
The paper presents an approach that leverages vision-language foundation models to transcribe and automate scientific experiments, particularly in chemistry and material science laboratories. By addressing challenges in lab digitization, this work demonstrates a promising method to enhance reproducibility and collaboration in scientific research.

**Strengths:**

1. **Novel Framework:** The paper introduces an innovative framework for automating data capture during laboratory experiments. The use of multi-modal transformer models to interpret high-dimensional raw data streams offers a fresh perspective on enhancing digitization in labs.

2. **Detailed Analysis:** The authors provide a comprehensive understanding of the challenges in lab digitization and how multi-modal learning can contribute to addressing these challenges.

3. **Successful Implementation:** The case study, which focuses on semiconductor research and wet chemistry, showcases the potential of the proposed framework. The results, wherein foundation models were fine-tuned on a limited dataset to generate reports with high similarity to recorded procedures, are commendable.

4. **Promising Research Direction:** The exploration of multi-modal models in automating experiment documentation has great potential, as echoed by both reviewers.

5. **Clear Writing:** The clarity of writing and presentation makes the paper accessible to readers, even those who might not be domain experts.

**Areas of Improvement for Acceptance as a Poster:**

1. **Introduction and Motivation:** A concise introductory section highlighting the motivation and primary results can help in setting the context right from the beginning. This will aid in better comprehending the paper's objectives and achievements.

2. **Computational Demands:** The paper could benefit from a section detailing the computational resources required for the multi-modal transformer's training process, ensuring readers have a clear understanding of its feasibility.

3. **Framework's Universality:** The paper should address the potential adaptability of the framework across different laboratory environments and other scientific domains.

4. **Foundation Model Emphasis:** The paper's emphasis on foundation models may seem overstated since the main contribution is fine-tuning a pre-trained network. Clarifying the actual contribution can help in setting the right expectations.

5. **Scalability Concerns:** Addressing the potential scalability challenges, especially as experimental procedures become more complex, would provide a more rounded view of the solution's applicability.

**Conclusion and Recommendation:**
The paper introduces a significant advancement in the realm of scientific experimentation digitization. While it showcases a commendable approach using vision-language foundation models, certain improvements can enhance its impact. Given its potential, the paper is recommended for acceptance as a poster. This will allow the authors to gather feedback and potentially iterate on their work, ensuring a more robust solution.